# Identifying Important Nodes in Trip Networks and Investigating Their Determinants

**DOI:** 10.3390/e25060958

**Published:** 2023-06-20

**Authors:** Ze-Tao Li, Wei-Peng Nie, Shi-Min Cai, Zhi-Dan Zhao, Tao Zhou

**Affiliations:** 1Compleχ Lab, Big Data Research Center, University of Electronic Science and Technology of China, Chengdu 610054, China; 2Complexity Computation Laboratory, Department of Computer Science, School of Engineering, Shantou University, Shantou 515063, China; 3Key Laboratory of Intelligent Manufacturing Technology (Ministry of Education), Shantou University, Shantou 515063, China

**Keywords:** distance trip network, urban structure, travel pattern, centrality index, participation index

## Abstract

Describing travel patterns and identifying significant locations is a crucial area of research in transportation geography and social dynamics. Our study aims to contribute to this field by analyzing taxi trip data from Chengdu and New York City. Specifically, we investigate the probability density distribution of trip distance in each city, which enables us to construct long- and short-distance trip networks. To identify critical nodes within these networks, we employ the PageRank algorithm and categorize them using centrality and participation indices. Furthermore, we explore the factors that contribute to their influence and observe a clear hierarchical multi-centre structure in Chengdu’s trip networks, while no such phenomenon is evident in New York City’s. Our study provides insight into the impact of trip distance on important nodes within trip networks in both cities and serves as a reference for distinguishing between long and short taxi trips. Our findings also reveal substantial differences in network structures between the two cities, highlighting the nuanced relationship between network structure and socio-economic factors. Ultimately, our research sheds light on the underlying mechanisms shaping transportation networks in urban areas and offers valuable insights into urban planning and policy making.

## 1. Introduction

Urban systems, which consist of a complex array of transportation infrastructure, such as roads, railways, and bus routes, play a vital role in shaping the mobility patterns of urban residents. They are abstracted into a series of complex (urban) networks. Understanding the network structural properties of these systems is therefore of utmost importance for policymakers and researchers [1,2,3,4,5,6].

In recent years, a significant body of research has been devoted to studying human mobility. The movement of people in the physical space follows a distinct scaling law [7,8,9,10,11,12,13,14]. González et al. found that human trajectories display a high degree of spatial–temporal regularity, which contrasts with the randomness predicted by prevalent models. Each individual has a characteristic travel distance and a significant likelihood of returning to a few frequently visited locations [9]. Despite using only two models and banknote flow data, these findings remain significant and highlight the need for further investigation into the mechanisms behind human mobility. Song et al. used empirical data on human mobility, captured by mobile-phone traces to show that the predictions of the CTRW models are in systematic conflict with the empirical results. They introduced two principles that govern human trajectories, allowing us to build a statistically self-consistent microscopic model for individual human mobility [15]. Zhou et al. introduced a trip competition mechanism to empower a mobility model to estimate population fluxes in all spatial ranges [16]. Ding et al. used a dataset of Chinese hospitals to analyze patient flow patterns and investigate the effects of patient characteristics, hospital features, and geographical location on patient mobility. The study found that patient mobility has strong regional and geographic agglomeration tendencies that are related to patient characteristics such as age, gender, and disease type, as well as hospital level, size, and speciality features [17].

Despite extensive research, there is still much to be learned about the patterns of human mobility, particularly in the context of urban transportation. Investigating the mobility patterns of individuals within cities, also known as urban human mobility, holds significant implications for a wide range of disciplines, including path planning, urban planning, congestion management, and disease transmission [18,19,20,21,22,23]. As such, further exploration and analysis of this topic are of great importance in the academic community.

As an expeditious and practical mode of transportation, the taxi constitutes a fundamental constituent of urban transportation systems. It is noteworthy that the movements of taxis are influenced by both passengers and drivers, and as such, it provides an accurate depiction of the mobility patterns of passengers within various urban hotspots and residential areas. Additionally, taxi movements are also determined by drivers seeking to maximize their profits. Consequently, taxi movements offer a comprehensive representation of the mobility behaviours of both passengers and drivers. Previous studies have predominantly focused on analyzing taxi movements through temporal and travel distance perspectives to facilitate the comprehension and forecasting of passenger mobility [24,25,26,27,28,29,30,31,32]. Liang et al. found that, in contrast to most models observed in human mobility data, the displacement of taxis travelling in urban areas tends to follow an exponential rather than a power-law distribution [25]. Xia et al. quantitatively explored the patterns of human mobility on weekends and weekdays. Through logarithmic binning and data fitness, they calculated the Bayesian weights to select the best fitting distributions [33]. Perlman et al. used the Uber Movement dataset to explore travel patterns of people at different times, locations, and modes of transportation and the impact of these patterns on the city’s transportation system. The study found that people’s travel behaviour in the Miami metropolitan area is influenced by many factors, including traffic congestion, time, weather, holidays, and events [34].

Insufficient research has been conducted to establish a solid framework for distinguishing long- and short-distance trips of taxis and to scrutinize the structural characteristics of urban areas and human mobility patterns, based on important nodes within the networks of long- and short-distance travel. In order to address this research gap, we conducted a rigorous analysis of probability density distributions of travel distance using taxi track data from both Chengdu and New York City. Subsequently, the long- and short-distance trip networks were constructed for both cities using a threshold of 95%. We employed the PageRank algorithm to identify the most important nodes within these networks. Our study then delved into the impact of these nodes on travel distance and classified them based on the centrality index and participation index. Furthermore, we investigated the underlying factors that contributed to the importance of these nodes. Overall, our research provides a comprehensive analysis of the structural and socio-economic factors that impact transportation networks, thereby aiding policymakers in enhancing mobility and accessibility for urban residents. By identifying key drivers of transportation patterns, our findings can inform the development of targeted policies that promote sustainable and equitable urban transportation systems. This research contributes to a broader understanding of the complex interplay between transportation, society and urban development, and provides actionable insights for policymakers and urban planners.

## 2. Materials and Methods

### 2.1. Data Description

Our research team collaborated with the local government in Chengdu, China, to address traffic congestion issues. As part of this effort, we obtained a dataset containing the trajectories of 10,037 authorized taxis belonging to a specific taxi company in Chengdu. Each taxi’s recording interval in the original dataset was 10 s, with each data item containing anonymized vehicle ID, timestamp, latitude, longitude, and occupancy flag (1 or 0). The dataset covers a time period from 1 August to 30 August 2014, with latitude and longitude recorded at meter-level accuracy. The occupancy flag, a Boolean variable, indicates whether a taxi was occupied (with passengers) or unoccupied (without passengers) at the recorded time. Furthermore, we gathered data of approximately 12 million taxi trajectories from New York City, United States, spanning from 1 March to 30 March 2016. Each row of the taxi trajectory data contains information on time, distance, latitude, and longitude. Finally, we obtained the GIS data necessary to construct and analyze the Chengdu and NYC administrative map.

### 2.2. The Administrative Map

Chengdu, situated in southwestern China, serves as the capital city of Sichuan Province, spanning between longitude $102^{{}^{\circ}}54^{\prime }$ E∼$104^{{}^{\circ}}53^{\prime }$ E and latitude $30^{{}^{\circ}}05^{\prime }$ N∼$31^{{}^{\circ}}26^{\prime }$ N. As of the end of 2022, Chengdu comprises 12 municipal districts, 3 counties, and 5 county-level cities, encompassing a total area of 14,335 km${}^{2}$ and accommodating a resident population of 21.192 million. The areas under study, illustrated in Figure 1a, predominantly pertain to the districts marked by numbers, which constitute the city centre, whereas the remaining ones constitute the suburbs.

New York City (NYC), situated on the Atlantic coast in the southeastern region of New York State, is the most populous city in the United States. As depicted in Figure 1b, NYC comprises five boroughs: The Bronx, Brooklyn, Manhattan, Queens, and Staten Island, encompassing a total area of 1214.4 km${}^{2}$, of which 425 km${}^{2}$ consists of water bodies and 789 km${}^{2}$ of land. The resident population of NYC is estimated to be approximately 8.62 million. Our primary study area includes the aforementioned five boroughs.

### 2.3. Network Construction

We constructed a trip network by utilizing taxi rules and GIS data mentioned above. Firstly, we divided the administrative maps of Chengdu and NYC into multiple small square cells, representing the nodes of the trip network, using the ArcGIS API from Environmental Systems Research Institute, Inc. (Redlands, CA, USA). Each small cell is considered a grid with a size of 1 km${}^{2}$ [24,31,35]. To ensure the maximum size of each trip network, we divided Chengdu’s and NYC’s trip networks into approximately 12,300 and 960 small cells, respectively.

An example of constructing a trip network is provided in Figure 2. As illustrated in Figure 2a, four cells are labelled by numbers. Our objective was to examine the impact of different trip distances on nodal importance in the trip network. Accordingly, the onboard trajectory data was used to construct the trip network. Finally, we extracted the departure and arrival time, the latitude and longitude of the starting and ending location from each loaded trajectory data. Based on the Euclidean distance, the travel distance of passengers was calculated. However, trips with excessively long or short distances were considered invalid data as they conflicted with passengers’ interests. According to people’s daily experience, the valid travel distance was set within [0.5 km, 200 km] [36]. We noted that the trips within proper travel distance accounted for 99.9% of the total trips.

In Figure 2b, the taxi trajectories have starting and ending positions located in different cells according to their respective latitude and longitude. The directed edges in the network represent the directions of the taxi trajectories. Figure 2c displays the resulting trip network, where the weight of the edges between cells *i* and *j* is set to n×d if there are *n* trajectories from cell *i* to cell *j*, with each edge having a length of *d*. Additionally, if the starting and ending positions are located in the same cell, a self-loop will be created for that cell.

### 2.4. Important Node Indicators

In this subsection, we introduce our metrics for calculating nodal importance. The metrics include the measurements of the hub node [17,37] and the participation index of the hub node [38]. PageRank is a popular algorithm used to determine the importance of nodes in a network graph. In urban transportation networks, this algorithm can be used to identify which road segments or transportation hubs have the greatest impact on the overall transportation network. Essentially, the PageRank algorithm determines node weights by calculating the number of inbound links to a node and the relative weight of those links. In urban transportation networks, nodes may represent intersections, bus stops, subway stations, etc. By calculating the PageRank value of each node, we can determine which nodes are most important and therefore need to be prioritized for maintenance and improvement [39,40,41,42,43]. Consequently, we employ the PageRank algorithm to identify important nodes in a trip network. Specifically, the top 5% of nodes identified through the PageRank algorithm are regarded as important ones [44], collectively constituting the cluster of the important nodes. Subsequently, we analyse this cluster, wherein the strength of the connections between a node *i* and other nodes in its cluster is measured using the variable $I_{i}$. Similarly, the variable $E_{i}$ is used to measure the strength of connections between node *i* and nodes outside the cluster that do not fall under the category of important nodes. These measurements serve as critical indicators in evaluating the efficacy of the identified clusters of the important nodes.
(1)$$\left\{\begin{array}{cc} I_{i}=\frac{k_{i}-\bar{k}}{\sigma_{k}} & \\ E_{i}=\frac{m_{i}-\bar{m}}{\sigma_{m}} \end{array}\right.$$

Equation (Equation 1) presents the internal and external cluster centrality indices, $I_{i}$ and $E_{i}$, respectively. These indices are defined in terms of $k_{i}$, $m_{i}$, $\bar{k}$, $\sigma_{k}$, $\bar{m}$, and $\sigma_{m}$. Specifically, $k_{i}$ represents the total weight of links between node *i* and other nodes within the same cluster. Similarly, $m_{i}$ denotes the total weight of links between node *i* and nodes outside the cluster. Meanwhile, $\bar{k}$ and $\sigma_{k}$ stand for the mean and standard deviation of the link weights of all nodes inside the cluster, respectively; $\bar{m}$ and $\sigma_{m}$ represent the mean and standard deviation of the link weights of all nodes outside the cluster, respectively. It should be noted that in this context, the weight of links is equivalent to edge weights.

Next, we utilize two participation indices, namely $P_{I_{i}}$ and $P_{E_{i}}$, to depict the distribution of node connectivity within and outside the cluster, respectively. We introduce a participation vector $P_{i}$ to compute participation indices [38]. Specifically, we construct an intra-cluster participation vector $P_{I_{i}}$ from the cluster perspective. The vector elements denote the probability of connection between node *i* and other nodes within the cluster. Additionally, we construct an out-of-cluster participation vector $P_{E_{i}}$ from the perspective of the entire trip network to portray the connection probability between node *i* and nodes outside the cluster. For instance, if the connections of node *i* in a network are uniformly distributed among the three nodes within its cluster, $P_{I_{i}}$ will be represented as ($\frac{1}{3},\frac{1}{3},\frac{1}{3}$). Ultimately, the participation index is defined as Equation (Equation 2),
(2)$$P_{i}=1-\frac{m}{\sqrt{m-1}}\sigma \left(P_{i}\right),$$
where *m* indicates the number of nodes in the cluster (for $P_{I_{i}}$) or the number of nodes outside the cluster (for $P_{E_{i}}$).

Nodes that possess centrality indices ($E_{i}$ and $I_{i}$) greater than 2 are referred to as hubs. These nodes are further divided into three categories based on the extent of their connectivity within and outside of the cluster. Nodes having a participation index within the range of (0, 0.3] are categorized as exclusive hubs, as they exhibit limited connectivity to nodes both inside and outside of the cluster. Nodes possessing a participation index within the range of (0.3, 0.6] are categorized as inclusive hubs due to their relatively extensive connectivity. Finally, nodes possessing a participation index within the range of (0.6, 1] are categorized as extensive hubs, owing to their wide-ranging connectivity [17].

## 3. Results

In this section, we aim to provide a rigorous analysis of trip networks by exploring the distance probability density distributions. We also endeavour to comprehensively examine the basic information pertaining to both long- and short-distance trip networks. Additionally, we aim to investigate the impact of important nodes on the distance probability distributions. We calculate important nodes’ centrality and participation indexes and categorize the important nodes according to their influence. Finally, we have introduced two distinct variables, namely demographic and economic factors, to characterize the important nodes comprehensively.

### 3.1. Travelling Distance Distribution

In Figure 3a,b, we can observe the travelling distance probability density distributions in Chengdu and NYC, respectively. Exponential distributions are a good approximation of these distance probability density distributions of both cities. However, a stronger heterogeneity of the travel distances exists in Chengdu compared to that of NYC. In Figure 3c,d, the distance probability density distribution and the corresponding cumulative distribution of both cities are presented. According to the Annual Report on Urban Transport Development in China (2019), short-distance trips (less than 5 km) account for nearly 60% of total trips in Chinese cities. The modes of short-distance trips in the report include shared bikes, taxis, buses, etc. Accordingly, it can be argued that the proportion of taxi trips taken for short distances should be increased, as they fall within the scope of this study. This assertion aligns with the focus and objectives of an academic research article [9,45,46]. In Figure 3c, when Distance ≥ 19, P(Distance) remains largely constant and P(Distance ≤ 19) > 95%; in Figure 3d, when Distance ≥ 11, P(Distance) remains largely constant and P(Distance ≤ 11) > 95%. Therefore, Distance = 19 and Distance = 11 are adopted as the dividing points of the long- and short-trip networks in Chengdu and NYC, respectively. That is, in the Chengdu trip network, the trip trajectories with Distance ≤ 19 constitute the short-distance trip network, and the remaining trip trajectories constitute the long-distance trip network.

### 3.2. Comparison of the Trip Network Parameters

In this subsection, we present a comparative analysis of the basic characteristics of the Chengdu and NYC trip networks, as depicted in Table 1. We aim to highlight the salient features of these networks and draw meaningful insights for transportation planning and policy making.

Our analysis reveals several commonalities between the trip networks of these two cities. Firstly, both exhibit redundancy in the number of nodes and edges between the long- and short-distance trip networks. Additionally, their average clustering coefficients are considerably low, suggesting a lack of clustering or interconnectedness among nodes. Finally, the average weighted strength of the nodes in the short-distance trip network is higher, indicating a higher frequency of mundane trips than long-distance trips.

### 3.3. Impact of Important Nodes on Travelling Distance

In this subsection, we utilize the PageRank algorithm to identify important nodes in the long- and short-distance trip networks of each city. Specifically, we consider the top 5% of nodes identified by the PageRank algorithm to be important ones [44]. Figure 4 depicts the impact of removing these important nodes from the network on the distance probability density distribution. Blue circles denote the distribution after removing the important nodes, while purple circles represent the original distribution before removal. Results for both the long- and short-distance trip networks in Chengdu and NYC are shown in Figure 4.

Our analysis reveals that the travelling distance probability density distribution of the Chengdu short-distance trip network conforms to the exponential distribution, regardless of whether the important nodes are removed or not (as evident from Figure 4a. However, Figure 4b indicates that the distance probability density distribution of the NYC short-distance trip network cannot be well modelled by the exponential form for the removal of important nodes. Furthermore, we find that both the long-distance trip networks in Chengdu and NYC are well approximated by the power-law distribution, even after removing the important nodes (as demonstrated in Figure 4c,d. It is noteworthy that the heterogeneity of travel distances is reduced to a certain degree after removing the important nodes. This highlights the significant impact of these nodes on urban travel. Thus, policymakers can enhance the transportation conditions and living environment near these critical nodes to improve the travel experience of city residents. Such efforts will likely lead to increased happiness among residents and further promote the development of these cities.

In summary, within long- and short-distance trip networks prior to the elimination of important nodes, the short-distance trip conforms to an exponential distribution. In contrast, the long-distance trip conforms to a power-law distribution [9,47,48]. This finding further validates our utilization of a 95% threshold to distinguish between long- and short-distance trip networks.

### 3.4. Classification of Important Nodes and Analysis of Influencing Factors

In this subsection, we present a calculation of the participation indices ($P_{I_{i}}$ and $P_{E_{i}}$) and cluster centre index ($I_{i}$ and $E_{i}$) for the long- and short-distance trip networks in NYC and Chengdu. To facilitate clear representation, we utilize the following notations: NYC_Long_Zexlusive to refer to exclusive nodes identified in the NYC long-distance trip network based on the cluster internal centre index $I_{i}$ and the cluster internal participation indices $P_{I_{i}}$, and NYC_Long_Bexlusive to refer to exclusive nodes identified in the same network based on the cluster external centre index $E_{i}$ and the cluster external participation indices $P_{E_{i}}$.

Figure 5 illustrates the distribution of hub nodes within the long- and short-distance trip network of Chengdu. Specifically, as depicted in Figure 5a, only a small number of nodes are tightly connected to the internal portion of the cluster. This is primarily due to the majority of nodes within the cluster being distributed across the five central districts of Chengdu and the long-distance trip network having a travel distance exceeding 19 km. Consequently, only a few nodes emerge as hub nodes that maintain relatively close connections with the interior of the cluster. In contrast, Figure 5c demonstrates that a substantial number of nodes serve as extensive hubs with robust connections to the external portion of the cluster. This finding suggests that nodes within the important node cluster of the long-distance trip network are more closely linked with nodes outside the cluster. In other words, taxi passengers tend to choose to travel outside the important node cluster (non-city centre) and inside the important node cluster (city centre) when travelling long distances. The findings presented in this study are illustrated in Figure 5b,d, which demonstrate that a significant number of nodes in the short-distance trip network serve as extensive hubs with tightly interconnected nodes both inside and outside the cluster. Specifically, this suggests that nodes within the cluster of important nodes, located in the city centre, exhibit high levels of connectivity to internal and external nodes. Consequently, it is reasonable to assume that when passengers opt for short-distance travel, they may choose to travel inside the important node cluster (city centre) or choose to travel between the important node cluster inside (city centre) and outside the cluster (non-city centre).

Figure 6 displays the distribution of hub nodes in the long- and short-distance trip networks of NYC. As depicted in Figure 6a, a very limited number of nodes act as inclusive hubs and extensive hubs in NYC’s long-distance trip network. These hubs are closely linked to nodes located within important node clusters. In contrast to Chengdu, NYC’s long-distance trip network contains extensive hubs, which suggests a higher level of interconnectedness between nodes within important node clusters. Additionally, Figure 6c shows that, similar to Chengdu, many nodes serve as extensive hubs that are closely connected to nodes outside the cluster. This finding indicates that nodes within important node clusters in the long-distance trip network exhibit a higher degree of connectivity to nodes located outside the cluster. As illustrated in Figure 6b,c, a substantial number of nodes serve as extensive hubs, possessing strong connections to nodes both inside and outside the important node cluster. These findings suggest that in short-distance trip networks, taxi passengers tend to travel within the cluster of important nodes, as well as between the important node cluster and areas outside it. Further analysis of Figure 6b–d reveals that certain nodes within the Manhattan district exhibit remarkable stability and serve as extensive hubs in both the short- and long-distance trip networks, indicating their significant influence within the trip network.

These results shed light on the underlying mechanisms of node clustering and connectivity within the long- and short-distance trip network. To elucidate the characteristics of important node cluster distributions, this study incorporated demographic and economic factors into the analysis. Note that the demographic and economic data used in the multivariate linear analysis are the years corresponding to the travel data. The demographic data utilized in our study were obtained from the WorldPop platform, which is an open statistics site. The demographic data we used corresponds to the resolution of the trip network, meaning that the WorldPop platform provides the number of people within each square kilometer. The economic data was obtained from the Bureau of Statistics of Chengdu, Sichuan Province, China and the U.S. Bureau of Statistics, respectively. Additionally, the economic data utilized in our study were sourced from the GDP data of each district within the city. Specifically, we employed multiple linear regression to examine the relationship between the node participation index and population and economy. This approach allowed for a more thorough investigation of the factors influencing the distribution of important node clusters.

In the context of the NYC long-distance trip network, our analysis reveals a significant linear relationship between the internal participation index of node clusters and both population and economy ($R^{2}$ = 0.735, Prob (F-statistic) = 9.14 $\times \;10^{-5}$). Notably, the economy exerts the greatest influence on this relationship (P>|t| = 0.002). Similarly, we observe a more pronounced linear relationship between the external participation index of node clusters and population and economy ($R^{2}$ = 0.302, Prob (F-statistic) = 0.00186), with the population having the strongest impact (P>|t| = 0.032).

Turning our attention to the NYC short-distance trip network, our findings indicate a significant linear relationship between the internal participation index of node clusters and both population and economy ($R^{2}$ = 0.565, Prob (F-statistic) = 1.07 $\times \;10^{-6}$), with the population having the most significant effect (P>|t| = 0.001). However, in contrast to the long-distance trip network, we do not observe a strong linear relationship between the external participation index of node clusters and population and economy.

Finally, in Chengdu’s long-distance trip network, a noteworthy linear correlation was observed between the internal participation index of node clusters and the population. The coefficient of determination ($R^{2}$) for this relationship is calculated to be 0.906, indicating a high degree of explanatory power. Notably, the following nodes were identified as having a significant impact on this relationship: Shuangliu Airport, Chengdu Airport Passenger Terminal in Shuangliu District, Century City in Wuhou District, Chengdu East Station, and Chengdu East Passenger Terminal. These findings provide valuable insights into the factors that drive the formation and evolution of long-distance travel networks in urban areas.

## 4. Discussion and Conclusions

In this study, our first step was to conduct an analysis of the probability density distributions for travel distances in Chengdu and NYC. Our findings revealed that the exponential distribution provided a good fit for both cities, with Chengdu exhibiting stronger heterogeneity. Subsequently, we utilized a threshold of 95% to divide the trip network into long- and short-distance trip networks. By analyzing these networks, we discovered that the power-law distribution accurately described long travel distances in both cities, while the exponential distribution provided a good fit for short travel distances [9,25,45,46,47,48]. These results further confirm the validity of our approach in using a 95% threshold to divide trip networks.

Second, the PageRank algorithm was applied to extract significant nodes in the trip networks of two cities, with a focus on academic research. Specifically, the top 5% of PageRank algorithm results were identified as important nodes [44], and their impact on the respective trip networks was analyzed. The findings suggest that the travel distance distribution in Chengdu remained unaltered, even after the removal of important nodes. In the context of the NYC short-distance trip network, the distribution of travel distances was observed to deviate from the exponential distribution after the removal of important nodes. Conversely, in the NYC long-distance trip network, travel distances were found to adhere well to the power-law distribution, regardless of the removal of important nodes. Additionally, in the short-distance trip network of the two cities, a noticeable reduction in the heterogeneity of travel distances was observed upon the removal of important nodes. These findings suggest that the impact of node removal on travel distance distribution varies across different types of trip networks.

The last stage of this research entailed categorizing the notable nodes based on their centrality and participation indices [17,37,38], and scrutinizing these classified nodes to investigate their origins. Specifically, we compared the hub nodes in the long-distance trip networks of the two cities and observed that the majority of the nodes in the important node cluster exhibited fewer interconnections with one another. However, nodes within the important node cluster were more closely linked with nodes outside the cluster. Conversely, within the short-distance trip network, the nodes in the cluster demonstrated a high level of connectivity, and the nodes within the cluster were also well-connected to those outside of it. These findings shed light on the structural differences in the two types of networks, and suggest that further exploration of the identified nodes may yield valuable insights into the underlying mechanisms governing the formation and operation of these networks [49,50,51]. In the final analysis, we incorporated demographic and economic variables to evaluate the important nodes. Our findings indicate that in the long-distance trip network of NYC, there exists a significant linear correlation between the participation index of nodes and both the economy and population. Meanwhile, the economy emerges as the dominant factor influencing the intra-cluster participation index of nodes. While the population serves as the dominant factor influencing the extra-cluster participation index of nodes. In contrast, within the NYC short-distance trip network, the intra-cluster participation index of nodes exhibits a significant linear association with both economy and population, with the population exerting the strongest influence. These results highlight the nuanced relationships between the network structure and socio-economic factors, and have implications for understanding the underlying mechanisms shaping transportation networks in urban areas [52,53,54,55].

Overall, taxis play a crucial role in complementing other modes of public transportation in Chengdu and NYC. While buses and metros have fixed stops, taxi flows can effectively identify areas where traffic demand exceeds current service levels. By modifying bus routes and extending metro lines, particularly to local centres, future traffic demands can be met, and accessibility can be improved. Our study provides valuable insights into the travel patterns of these two cities and highlights the usefulness of statistical models in analyzing such patterns. Furthermore, this research sheds light on the underlying mechanisms that shape transportation networks in urban areas, which has important implications for urban planning and policy making. By understanding the structural and socio-economic factors that influence transportation networks, policymakers can work to improve mobility and accessibility for urban residents. These observations have important implications for transportation planning in these cities. For instance, policymakers could focus on incentivizing long-distance travel to promote tourism and economic growth.

We contend that the present study and its methodology hold promise for investigating other facets of transportation networks. However, it is important to acknowledge the limitations of using taxi data as a representative sample. Firstly, taxi trips can only provide insights into a portion of intra-urban travel. Secondly, the temporal coverage of the dataset is relatively limited, which may lead to some degree of bias in the results obtained. Clearly, different transportation modes and varying characteristics of human mobility may reveal diverse aspects of urban structure. As such, future research could benefit from utilizing a dataset that encompasses several transportation modes and spans a longer time period, in order to gain a more comprehensive understanding of important nodes in urban areas and the factors that contribute to their emergence.

## Figures and Tables

**Figure 1 entropy-25-00958-f001:**
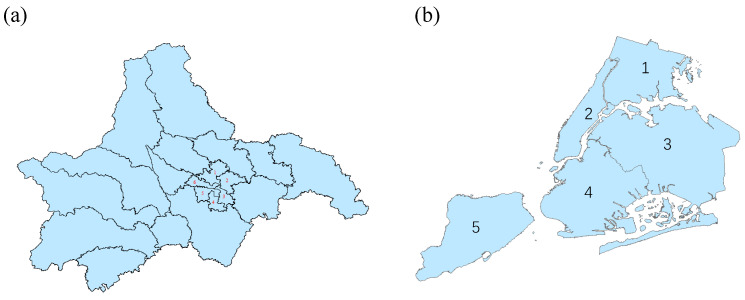
(**a**) The administrative map of Chengdu. It divides Chengdu into two regions: the central and suburban areas. The former encompasses six districts, labelled with red numbers, which are 1: Jinniu District; 2: Chenghua District; 3: Jinjiang District; 4: Gaoxin District; 5: Wuhou District; and 6: Qingyang District. The latter includes the remaining districts. (**b**) The administrative map of NYC. It divides NYC into five distinct regions, with the main area comprising five districts marked by black numbers. These districts are 1: The Bronx; 2: Manhattan; 3: Queens; 4: Brooklyn; and 5: Staten Island.

**Figure 2 entropy-25-00958-f002:**
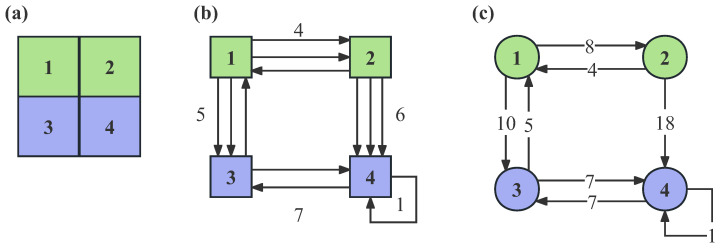
Schematic diagram of constructing the trip network. (**a**) The division of the administrative map into cells. (**b**) The connections between cells according to taxi trajectories. (**c**) The resulting trip network generated by taxi trajectories.

**Figure 3 entropy-25-00958-f003:**
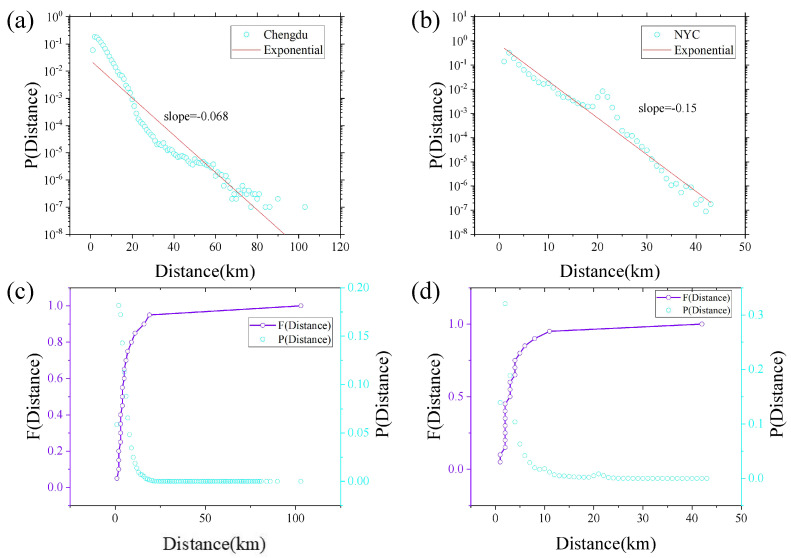
(**a**,**b**) The linear–logarithmic plot of the distance probability density distribution of the Chengdu and NYC trip network. The horizontal axis is the distance, and the vertical axis is the corresponding probability. The red line indicates the results of fitting the probability density distribution. (**c**,**d**) the linear–linear plots of the cumulative distribution for the Chengdu and NYC trip network.

**Figure 4 entropy-25-00958-f004:**
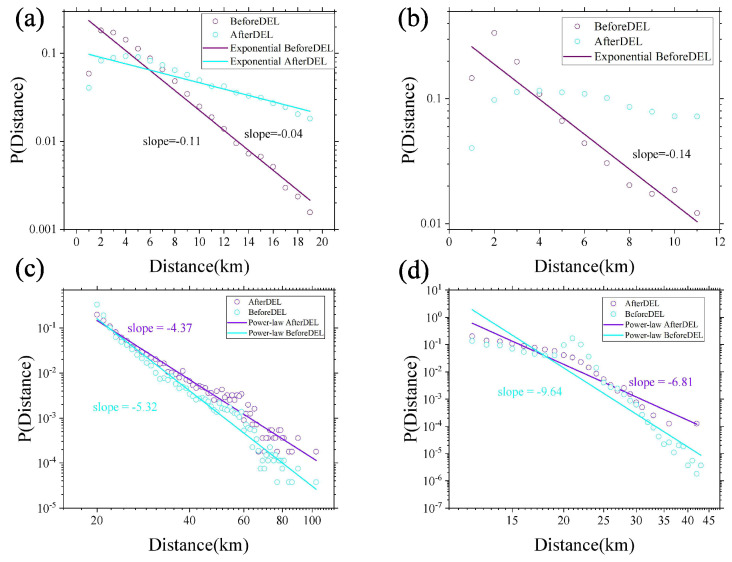
(**a**,**b**) The linear–logarithmic plots of the distance probability density distribution of the Chengdu and NYC short-distance trip network, respectively. The horizontal axis is the trip distance. The vertical axis is the corresponding probability. The results of fitting the probability density distribution are the colourful lines. (**c**,**d**) the logarithmic–logarithmic plots of the distance probability density distribution of the Chengdu and NYC long-distance trip networks, respectively. The horizontal axis is the trip distance. The vertical axis is the corresponding probability. The results of fitting the probability density distribution are the coloured lines.

**Figure 5 entropy-25-00958-f005:**
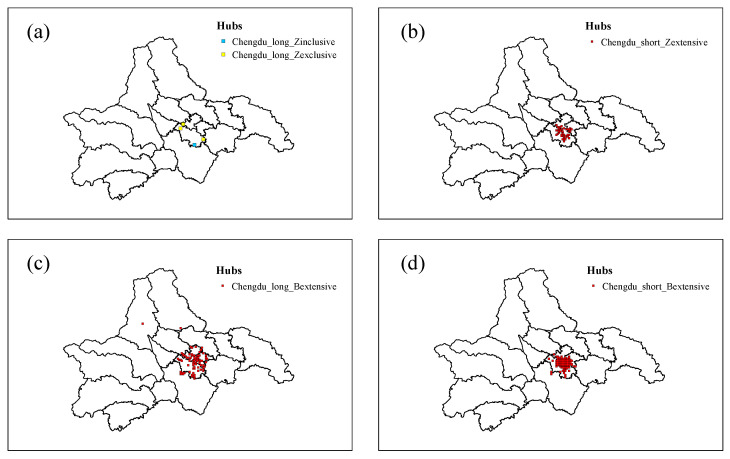
Distribution of hub nodes in Chengdu long- and short-distance trip network. (**a**) Distribution of internal inclusive hubs (Chengdu_long_Zinclusive) and internal exclusive hubs (Chengdu_long_Zexclusive) in the Chengdu long-distance trip network. (**b**) Distribution of internal extensive hubs (Chengdu_short_Zextensive) in the Chengdu short-distance trip network. (**c**) Distribution of external extensive hubs (Chengdu_long_Bextensive) in the Chengdu long-distance trip network. (**d**) Distribution of external extensive hubs (Chengdu_short_Bextensive) in the Chengdu short-distance trip network.

**Figure 6 entropy-25-00958-f006:**
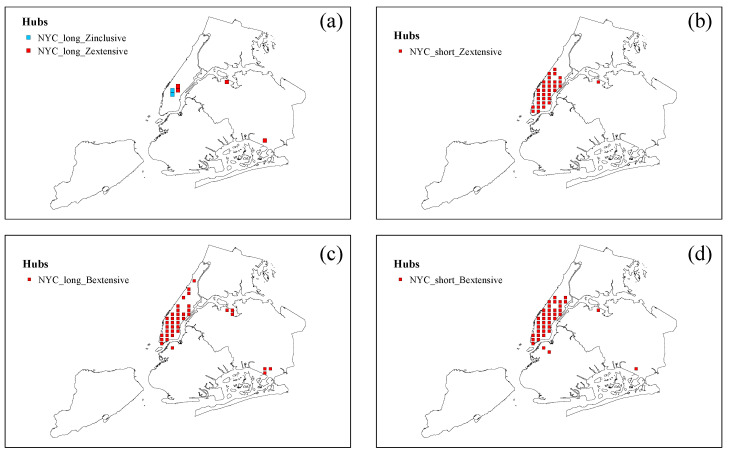
Distribution of hub nodes in NYC long- and short-distance trip network. (**a**) Distribution of internal inclusive hubs (NYC_long_Zinclusive) and internal extensive hubs (NYC_long_Zextensive) in the NYC long-distance trip network. (**b**) Distribution of internal extensive hubs (NYC_short_Zextensive) in the NYC short-distance trip network. (**c**) Distribution of external extensive hubs (NYC_long_Bextensive) in the NYC long-distance trip network. (**d**) Distribution of external extensive hubs (NYC_short_Bextensive) in the NYC short-distance trip network.

**Table 1 entropy-25-00958-t001:** Comparison of the distance trip network parameters.

	Chengdu		NYC	
**Type of Network**	**Short**	**Long**	**Short**	**Long**
Number of Nodes	2394	2510	780	859
Number of Edges	169,372	12,966	38,454	24,642
Average Weighted Strength of Node	18,280.40	302.30	39,787.00	12,343.60
Average Shortest Path Length	17.93	49.79	12.38	23.47
Average Clustering Coefficient	$7.0\times 10^{4}$	$8.0\times 10^{6}$	$6.0\times 10^{4}$	$5.0\times 10^{4}$
Assortativity Coefficient	−1.06 × 10^1^	−4.20 × 10^2^	$4.00\times 10^{3}$	−8.50 × 10^−2^

## Data Availability

Data will be made available on request.

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
