# Peer review of "Identifying Important Nodes in Trip Networks and Investigating Their Determinants"

_entropy, 2023, doi:10.3390/e25060958_

Round 1
Reviewer 1 Report
In this paper, the authors analyze the taxi trip data and construct the trip networks to study the important nodes and their characteristics. In general, the work is interesting and explicitly indicate the structural features of the trip networks in two cities. I have the following questions before it can be accepted:
1. The authors indicate that one of the objective is to examine the impact of different trip distances on nodal importance in the trip network. But as far as I’m concerned, the nodal importance is not determined by the trip distances, but by the geographical location of the node in the city.
2. In line 168, the hubs are those ‘with centrality indices greater than 2’. What centrality index is referred to here? The I_i, E_i or PageRank? Is there any reason that you use the index to identify hubs? The groups of nodes with participation index ranging in (0,0.3 ] and (0.6,1] are categorized as exclusive hubs, but their connectivity patterns are different. Why not distinguish them?
3. The contribution of the paper is not very focused. There are many points proposed by the author, such as constructing and analyzing the structure of trip network, identify and classify the important nodes, analyze the distance probability density distributions, introduce variables to characterize the important nodes. I suggest the authors to emphasize on their key contributions.
4. In line 292, the authors declare a linear relationship between the internal participation index of nodes and the population and economy, but these two indexes are abstractive here. The authors should explain the specific quantities of population and economy that are used in calculation.
Reviewer 2 Report
“Our findings indicate that in the long-distance trip network of NYC, there exists a significant linear correlation between the participation index of nodes and both the economy and population. Meanwhile, the economy emerges as the dominant factor influencing the intra-cluster participation index of nodes. While the population serves as the dominant factor influencing the extra-cluster participation index of nodes.“
I don't see the adequacy of this conclusion. There are no figures and results that support this assertion.
The overall quality of the paper is quite suitable for publication.
I would like to see more connections between the trip network structures and soc-economic factors.
Reviewer 3 Report
The paper presents a methodology for firstly, identifying important nodes/locations in cities based on large data sets on urban taxi trips and secondly, for investigating which factors drive the emergence of important nodes.
The paper is mostly very well written, with only a few typos (e.g., p2 line 59), and some mixing of past/present tense.
The computation and calculations seem solid, and figures are mostly clear and easy to understand.
## Abstract
The abstract is well written in terms of describing the paper content, but it does not sufficiently describe what research problem/knowledge gap the paper is trying to solve.
## Introduction
The manuscript states that there still is "much to be learned" about patterns of human mobility but does not specify *what* there is to be learned, or how the paper is related to any identified knowledge gap.
Given the framing of the paper (relevance for planners a policy makers) the paper lacks a more precise conceptualization of both what *research problem* the paper is addressing and, ideally, also how that tie to more *practical issues* within urban mobility planning etc.
## Materials and methods
### PageRank Algorithm
Given that the PageRank algorithm has been developed for a completely different problem than urban mobility - and rarely is used for transport networks - the manuscript is missing a motivation for why this way of measuring centrality node centrality is the most suitable, and potentially also how it influences the outcome compared to more conventional choices for measuring node centrality in transportation networks.
### Data
The dataset only contains taxi trips from a specific company and from quite short time spans. The manuscript is missing some discussion of how representative taxi data are in general of urban mobility (this is only briefly hinted at in the final discussion), as well as these data specifically. Are they for example representative of taxi/trip data in the two cities generally? Especially given that the paper states that the Chinese taxi data's trip distribution does not match the trip distribution reported by official sources (p5 194-195).
Would the method also work for other types of mobility data?
## Methods
The description of how the taxi trips is converted to a network structure is clear and easy to follow, but I have some concerns about how the data modeling influences the results.
Some quite significant choices are made in the network modeling, such as the distance of the grid cells, their exact placement, and the choice to use Euclidean rather than street network distances. Based on the current descriptions, it is unclear whether any sensitivity analysis has been made, and if using e.g., a resolution has any effect on the main conclusions (for example by looking into the 'modifiable areal unit problem')
Given the vastly different cities the paper is based on in terms of street network structure, size, and population distribution, it would also be beneficial with some discussion of how e.g., grid resolution relates to city size.
Since the topic of the paper is the geographic distribution of taxi trips, some further work on how the geographies of the two cities influence the observer patterns.
## Results
The issue of network modeling means that the findings about redundancies and clustering of nodes and edges (p.6,210-212) are hard to interpret/relate to actual trip patterns, given that both node and edges are highly aggregated and presumably very sensitive to how the taxi data have been converted to a network structure.
The results section could generally use some additional meta-comments about the findings, detailing to what extent results are expected/unexpected or different from related, previous research, as well as what the implications are for understandings of human mobility patterns. For example, that long-distance trips tend to connect the non-central locations with the city center is both expected and in line with previous research. Similar, the results that population is correlated to high activity clusters, and that airports and major transport hubs generate more trips, are expected (p10,305-311), so some further elaboration of how the method/insights can provide new insights would be an improvement.
Adding context to why the two cities produce different results would make the findings more relevant generally.
For section 3.3. on removing important nodes, it is unclear 1) why comparing results with and without important nodes is insightful and 2) what the implications are for the differing results of this step for the two cities?
For the section on population and economy, details on how these two variables have been examined (data, resolution etc.) are as far as I can see completely missing.
The terminology 'NYC_Long_Zexlusive' etc. can be slightly confusing - it might be useful to include a table explaining the exact meaning of all the terms?
## Discussions and Conclusions
The discussion point to important limitations in using taxi data as a proxy for urban mobility. For this reason, I don't think the conclusion that the findings could inspire efforts to "improve the clustering level of the networks" are meaningful (p.11,366-367): given that this shows the connectivity of taxi trips and not the connectivity of the actual physical mobility networks, a lack of interconnectivity among nodes does not mean that the locations are not actually connected, and it is unclear why interconnectivity among taxi trips is something to optimize for?
Round 2
Reviewer 1 Report
The questions proposed are addressed and explained.
Author Response
Thank you again for your comments. These comments have been very helpful in our research.
Reviewer 3 Report
Thank you for considering the feedback. The new abstract gives a better understanding of the aim of the paper.
Regarding your response to question 3, 4 and 5, these responses should ideally be incorporated into the paper itself, to also be of use of other, future readers.
In terms of the new explanation of the population and economy variables, some further clarification on the data have been used would still be beneficial and make the method much clearer (e.g. spatial resolution, type of economic statistics, how data were connected to the network, etc.). If the word limit on the paper does not allow this, it could be in supplementary materials?
The cover letter and revised manuscript do, as far as I can tell, not address any of the comments for "Methods", "Results" (other than the economy/population variables) or "Discussions and conclusions":
## Methods
The description of how the taxi trips is converted to a network structure is clear and easy to follow, but I have some concerns about how the data modeling influences the results.
Some quite significant choices are made in the network modeling, such as the distance of the grid cells, their exact placement, and the choice to use Euclidean rather than street network distances. Based on the current descriptions, it is unclear whether any sensitivity analysis has been made, and if using e.g., a resolution has any effect on the main conclusions (for example by looking into the 'modifiable areal unit problem')
Given the vastly different cities the paper is based on in terms of street network structure, size, and population distribution, it would also be beneficial with some discussion of how e.g., grid resolution relates to city size.
Since the topic of the paper is the geographic distribution of taxi trips, some further work on how the geographies of the two cities influence the observer patterns.
## Results
The issue of network modeling means that the findings about redundancies and clustering of nodes and edges (p.6,210-212) are hard to interpret/relate to actual trip patterns, given that both node and edges are highly aggregated and presumably very sensitive to how the taxi data have been converted to a network structure.
The results section could generally use some additional meta-comments about the findings, detailing to what extent results are expected/unexpected or different from related, previous research, as well as what the implications are for understandings of human mobility patterns. For example, that long-distance trips tend to connect the non-central locations with the city center is both expected and in line with previous research. Similar, the results that population is correlated to high activity clusters, and that airports and major transport hubs generate more trips, are expected (p10,305-311), so some further elaboration of how the method/insights can provide new insights would be an improvement.
Adding context to why the two cities produce different results would make the findings more relevant generally.
For section 3.3. on removing important nodes, it is unclear 1) why comparing results with and without important nodes is insightful and 2) what the implications are for the differing results of this step for the two cities?
For the section on population and economy, details on how these two variables have been examined (data, resolution etc.) are as far as I can see completely missing.
The terminology 'NYC_Long_Zexlusive' etc. can be slightly confusing - it might be useful to include a table explaining the exact meaning of all the terms?
## Discussions and Conclusions
The discussion point to important limitations in using taxi data as a proxy for urban mobility. For this reason, I don't think the conclusion that the findings could inspire efforts to "improve the clustering level of the networks" are meaningful (p.11,366-367): given that this shows the connectivity of taxi trips and not the connectivity of the actual physical mobility networks, a lack of interconnectivity among nodes does not mean that the locations are not actually connected, and it is unclear why interconnectivity among taxi trips is something to optimize for?
Author Response
Thank you for your comments. Please see attached for detailed response.

Round 3
Reviewer 3 Report
Thank you for considering the feedback. I believe the paper is improved substantially after the second round of revisions, and it is easier to follow your methodology.